# Baseline Evaluation of the Impact of Updates to the MIT Earth System Model on its Model Parameter Estimates

Alex G. Libardoni[1], Chris E. Forest[1,2], Andrei P. Sokolov[3], and Erwan Monier[3]

[1]Department of Meteorology, Pennsylvania State University, University Park, Pennsylvania, USA
[2]Earth and Environmental Systems Institute, Pennsylvania State University, University Park, Pennsylvania, USA
[3]Joint Program on the Science and Policy of Global Change, Massachusetts Institute of Technology, Cambridge, Massachusetts, USA

*Correspondence to:* Chris E. Forest (ceforest@psu.edu)

**Abstract.** For over twenty years, the Massachusetts Institute of Technology Earth System Model (MESM) has been used extensively for climate change research. The model is under continuous development with components being added and updated. To provide transparency in the model development, we perform a baseline evaluation by comparing model behavior and properties in the newest version to the previous model version. In particular, changes resulting from updates to the land surface model component and the input forcings used in historical simulations of climate change are investigated. We run an 1800-member ensemble of MESM historical climate simulations where the model parameters that set climate sensitivity, the rate of ocean heat uptake, and the net anthropogenic aerosol forcing are systematically varied. By comparing model output to observed patterns of surface temperature changes and the linear trend in the increase in ocean heat content, we derive probability distributions for the three model parameters. Furthermore, we run a 372-member ensemble of transient climate simulations where all model forcings are fixed and carbon dioxide concentrations are increased at the rate of 1% per year. From these runs, we derive response surfaces for transient climate response and thermosteric sea level rise as a function of climate sensitivity and ocean heat uptake. We show that the probability distributions shift towards higher climate sensitivities and weaker aerosol forcing when using the new model and that the climate response surfaces are relatively unchanged between model versions. Because the response surfaces are independent of the changes to the model forcings and similar between model versions with different land surface models, we suggest that the change in land surface model has limited impact on the temperature evolution in the model. Thus, we attribute the shifts in parameter estimates to the updated model forcings.

## 1 Introduction

Equilibrium climate sensitivity (ECS), the equilibrium global-mean surface temperature change due to a doubling of atmospheric carbon dioxide concentrations, is a climate system property that has been widely studied and strongly influences future climate projections. One of the complexities of ECS is that it is a function of many feedbacks and processes that act on different spatial and temporal scales. In particular, the lapse rate, water vapor, cryosphere, and cloud feedbacks play especially critical roles in determining the climate sensitivity (Bony et al., 2006). Given its influence on future climate change, many studies using a range of methods have attempted to estimate ECS.

One class of studies estimates ECS directly from observations using a global energy budget approach (Gregory et al., 2002; Otto et al., 2013; Lewis and Curry, 2014; Masters, 2014). These studies calculate probability distributions of ECS from estimates of global mean surface temperature change, the heat stored in the ocean, and changes in radiative forcing, along with the associated uncertainties in their measurements. A second class of studies use simplified climate models such as Earth system models of intermediate complexity (EMICs) or energy balance models (e.g., Forest et al., 2002; Knutti et al., 2003; Forest et al., 2008; Libardoni and Forest, 2013; Olson et al., 2013; Johansson et al., 2015). Taking advantage of the computational efficiency of the simplified models, these studies run large ensembles over a range of climate sensitivity values in addition to adjusting other relevant factors, such as the rate of ocean heat uptake and a measure of the net aerosol forcing. By comparing model runs to observations and evaluating how well individual model runs match the past, estimates of ECS and other parameters are given as probability distributions.

Transient climate response (TCR) provides a second metric for estimating future climate change and is defined as the global mean surface temperature change at the time of carbon dioxide ($CO_2$) doubling in response to $CO_2$ concentrations increasing at the rate of 1% per year. $CO_2$ doubling occurs in year 70 of this scenario, making TCR a shorter-term assessment of climate change than ECS. Unlike ECS, which requires reaching an equilibrium state, TCR is estimated while the climate system is still adjusting to a time-dependent forcing. There is a constant evolution in the strength and activity of processes and feedbacks in both the atmosphere and the ocean as the climate system adjusts to reach equilibrium. Due to the long time scales required to reach equilibrium, Allen and Frame (2007) argue that we should focus on estimating TCR, which is more policy-relevant than ECS. Estimates of TCR can be made from current historical observations and are more meaningful on the decadal time scale, whereas even if the equilibrium response is known, it may never be reached. However, even if more focus is placed on TCR than ECS, the two are closely linked. When considering atmosphere-ocean interactions, TCR has been shown to depend on both climate sensitivity and the rate at which heat is mixed into the deep ocean (Sokolov et al., 2003; Andrews and Allen, 2008).

One EMIC that has been extensively used in studies estimating ECS and TCR is the climate component of the Massachusetts Institute of Technology (MIT) Integrated Global Systems Model (IGSM, Sokolov et al., 2005). Forest et al. (2002, 2006, 2008) and Libardoni and Forest (2011, 2013) estimated the joint probability distribution for climate sensitivity and other model parameters in IGSM. Each study used similar, but not identical, versions of IGSM with changes both to key components of the model and to the input data used to force the model. Climate change diagnostics were also modified in the studies. The Earth system component of IGSM has undergone further development and a new, updated version incorporated into the integrated framework. This study serves as a baseline evaluation of how probability distributions for the model parameters change as a result of updating the Earth system component. More specifically, we investigate the impact of (1) the structural changes to the model, (2) the historical datasets used to force the model, and (3) the sampling strategy used to vary the model parameters.

In the past, "IGSM" has been used to reference both the fully integrated model as well as the standalone Earth system component. We follow this convention and refer to the older version of the Earth system model as IGSM, and we refer to the updated version of the model as the MIT Earth System Model (MESM). In this study, we provide a transparent method of testing and accounting for how the simulated behavior and probability distribution functions change in response to the recent

model development. We derive a new joint probability distribution by closely following the methods of Libardoni and Forest (2011) to show the impact that the new version of the model has on the parameter estimates and find that the new version of the model leads to higher climate sensitivity estimates in addition to shifts in the distributions of the other model parameters. The effects on the parameter distributions due to changing observations and temperature metrics will be addressed in future studies in order to separate their impacts from changes due to the model update alone. We also show here how the emergent behavior of MESM compares to the older IGSM by running a new set of transient simulations and calculating how the response surfaces for TCR and thermosteric sea level rise depend on ECS and the rate of ocean heat uptake.

In Sect. 2, we give a brief description of the MIT modeling framework and the differences between IGSM and MESM. We describe the process for deriving the joint probability distribution function used in Libardoni and Forest (2011) and the modifications implemented in this study in Sect. 3. Parameter distributions and response surfaces are presented in Sect. 4. In particular, we test whether changes in the distributions and responses are due to reducing the number of model diagnostics, the sampling of the parameter space, or changes in the model structure and input forcings. We present our conclusions in Sect. 5.

## 2 Model

The climate component of the updated MIT Earth System Model (Sokolov et al., 2018) replaces the version described in Sokolov et al. (2005) and is an Earth system model of intermediate complexity. It consists of a zonally-averaged atmosphere, zonally-averaged land model, and a mixed-layer anomaly diffusing ocean model. The mixed-layer ocean model includes specified vertically-integrated horizontal heat transport by the deep ocean, a so-called "Q-flux". This flux has been calculated from a simulation in which sea surface temperatures and the sea-ice distribution were relaxed toward their present-day climatology. Heat mixing into the deep ocean is parameterized by the diffusion of the difference of the temperature at the bottom of the seasonal thermocline from its value in a pre-industrial climate simulation (Hansen et al., 1984; Sokolov and Stone, 1998). Since this diffusion represents the cumulative effect of heat mixing by all physical processes, the values of the diffusion coefficients are significantly larger than those used in the sub-grid scale diffusion parameterizations in ocean global circulation models. The spatial distribution of the diffusion coefficients used in the diffusive model is based on observations of tritium mixing into the deep ocean (Hansen et al., 1988).

The radiation code takes into account major greenhouse gases ($H_2O$, $CO_2$, $CH_4$, $N_2O$, CFCs, and $O_3$) and multiple types of aerosols (e.g. $SO_2$, black and organic carbon). In historical climate simulations, loading for all aerosols except sulfate are kept at their default values. The forcing due to sulfate aerosol is parameterized through changes in surface albedo using historical data on $SO_2$ emissions. Historical climate simulations are initialized from conditions obtained from a long equilibrium simulation for 1860 conditions.

Three model parameters that impact the climate system response are easily modified in MESM. These parameters are the equilibrium climate sensitivity (ECS), the effective ocean diffusivity ($K_v$), and the net aerosol scaling factor ($F_{aer}$). ECS is changed by adjusting the strength of the cloud feedback at different levels in the model (Sokolov, 2006; Sokolov and Monier, 2012). The adjustment required for a specific ECS is obtained from a lookup table derived from model simulations with

different feedback strengths where $CO_2$ concentrations have been doubled and the climate system allowed to reach equilibrium. $K_v$ represents the global mean ocean diffusion coefficient in the mixed-layer ocean model. The global mean diffusivity is adjusted by scaling the spatial diffusivity pattern by the same factor at all locations. A lower global mean diffusivity implies slower mixing of heat into the deep ocean and a higher global mean diffusivity implies faster mixing. The albedo adjustment used for the sulfate aerosol forcing is prescribed by a latitude-dependent pattern that differs over land and ocean (Forest et al., 2001). This pattern is held fixed spatially but scaled temporally by estimated emissions of sulfur dioxide. $F_{aer}$ sets the amplitude of the pattern in the 1980s. By choosing a set of the three parameters, $\theta = (ECS, K_v, F_{aer})$, we simulate different climate states.

We now highlight two major updates made between the current version of MESM and its predecessor. The first update was the incorporation of a new land surface model. The Community Land Model (CLM) version 3.5 (Oleson et al., 2008) replaced CLM version 2.1 to improve estimates of the surface heat balance in the model. A second update to the model was an adjustment to the radiative forcing of non-$CO_2$ greenhouse gases in the radiation code. The adjustment was made to match the calculations used in the Intergovernmental Panel on Climate Change (IPCC) experiments and produces weaker forcing for those constituents. Additionally, the forcings used to drive the model (Forest et al., 2006) were extended and, in some cases, new data sources were used. Greenhouse gas concentrations and stratospheric aerosols from volcanic eruptions were obtained from the National Aeronautics and Space Administration Goddard Institute for Space Studies modeling group forcing suite. The procedure for updating the greenhouse gas emissions from Hansen et al. (2007) and the volcanic aerosol forcing from Sato et al. (1993) was described in Miller et al. (2014). Updates included incorporating data from more observational sources and extending the length of the datasets. Sulfate aerosol loading from Smith et al. (2011) was extended to 2011 by Klimont et al. (2013). The Kopp and Lean (2011) solar irradiance dataset replaced the Lean (2000) dataset. Lastly, the ozone concentration database developed by the Atmospheric Chemistry and Climate initiative (AC&C) and Stratospheric Processes and their Role in Climate project (SPARC) ozone concentration database (Cionni et al., 2011) that was developed in support of the Coupled Model Intercomparison Project phase 5 (CMIP5) replaced the concentration data used in Forest et al. (2006). The concentrations in the dataset, hereafter referred to as AC&C/SPARC, drive the tropospheric and stratospheric ozone forcing in the radiation code. In Sect. 4, we show the differences between the old and new datasets for those forcings where the data sources have changed, namely solar and ozone.

## 3   Methods

In this section, we present an outline of the methodology used to derive the joint probability distribution function (PDF) for the model parameters and highlight the changes implemented between this study and previous studies using IGSM. We follow closely the methods of Libardoni and Forest (2011), which we briefly summarize here. To derive the PDFs, we compare output from each model simulation to time series of observed climate change. A given model run is evaluated through the use of a goodness-of-fit statistic

$$r^2 = (\boldsymbol{x}(\theta) - \boldsymbol{y})^T \mathbf{C_N^{-1}}(\boldsymbol{x}(\theta) - \boldsymbol{y}), \tag{1}$$

where $\boldsymbol{x}(\theta)$ and $\boldsymbol{y}$ are vectors of model output for a given set of model parameters and observed data, respectively, and $\mathbf{C}_N^{-1}$ is the inverse of the noise-covariance matrix. In its simplest form, the $r^2$ statistic is the weighted sum of squares residual between the model simulation and the observed pattern. The weights applied to the residuals are estimated from the unforced climate variability in a fully coupled, three-dimensional model and represent the observed patterns we would expect in the absence

of external forcings. In Libardoni and Forest (2011), surface temperature, upper-air temperature, and global mean ocean heat content patterns were used to evaluate model performance. We note that the definition of $r^2$ presented here is different than the coefficient of determination for the goodness-of-fit of a linear model. In a linear model, high values of $r^2$ indicate a good fit to the model. In our weighted sum, low values of $r^2$ indicate a good fit between the model output and the observations.

The goodness-of-fit statistics for each pattern used to evaluate the model are converted to a PDF using the likelihood function

function described in Libardoni and Forest (2011) and modified by Lewis (2013). Through an application of Bayes' Theorem, the individual likelihoods are combined to derive a joint PDF for the three model parameters. As in Libardoni and Forest (2011), we apply an expert prior to ECS and uniform priors to $K_v$ and $F_{aer}$. Marginal probability distributions for individual parameters are calculated by integrating the joint PDF over the other two parameters.

We make two changes to the methodology of Libardoni and Forest (2011) to derive PDFs using MESM simulations. First,

we run the model for $\theta$s that sample individual parameters over a wider range and on a more regular grid. Climate sensitivity is sampled from 0.5 to 10.0 °C in increments of 0.5 °C by adjusting the strength of the cloud feedback, the square root of ocean diffusivity is sampled from 0 to 8 cm s$^{-1/2}$ in increments of 1 cm s$^{-1/2}$, and the aerosol forcing amplitude is sampled from -1.75 to 0.5 Wm$^{-2}$ in increments 0.25 Wm$^{-2}$. By choosing this sampling strategy, we have increased the number of runs from 640 with IGSM to 1800 runs with MESM, widened the range of parameter values sampled, and increased the density of model

runs within the parameter space (Fig. 1).

As a second change, we reduce the number of diagnostics used to evaluate model performance. In general, independent temperature patterns should be used to evaluate model performance because they rule out different regions of the parameter space for being inconsistent with the observed climate record. In particular, Urban and Keller (2009) show that surface temperature and ocean heat content time series provide good constraints on model estimation. Further, Lewis (2013) shows upper-air

temperatures to be highly correlated with surface temperature via the lapse rate and water vapor feedbacks. For these reasons, we now omit the upper-air temperature diagnostic. The removal of the upper-air diagnostic leaves two temperature diagnostics for evaluating model performance: (1) decadal mean surface air temperature anomalies from 1946-1995 with respect to a 1906-1995 climatology in four equal-area zonal bands, and (2) the linear trend in global mean ocean heat content from 1955-1995 in the 0-3 km layer. As in Libardoni and Forest (2011), we use five surface temperature datasets (Jones and Moberg, 2003; Brohan

et al., 2006; Smith et al., 2008; Hansen et al., 2010) and one ocean heat content dataset (Levitus et al., 2005) as observations. Five different joint PDFs are derived by combining the likelihood from the ocean diagnostic with the likelihood derived from each of the individual surface temperature datasets.

**Figure 1.** Parameter pairings where the models have been run. Points in black are common to both the IGSM and MESM ensembles. Blue points are unique to the IGSM ensemble and red points are unique to the MESM ensemble.

## 4   Results

Our results are presented as follows. We first identify the changes in the input forcings used in our historical simulations by comparing the solar and ozone components used in the IGSM runs with those used in the MESM runs. Second, we show how the probability distribution functions change when reducing the number of model diagnostics from three to two through the omission of the upper-air diagnostic. Third, we derive probability distributions using the MESM ensemble and directly compare them to those derived using the IGSM ensemble using the full ensembles and the case where only runs with $\theta$s common to both ensembles are used. Fourth, we evaluate how well the model captures the observations by comparing model output from the MESM ensemble to the observed climate record. Finally, we derive the response surfaces for transient climate response and thermosteric sea level rise for MESM and compare them to the corresponding surfaces from IGSM.

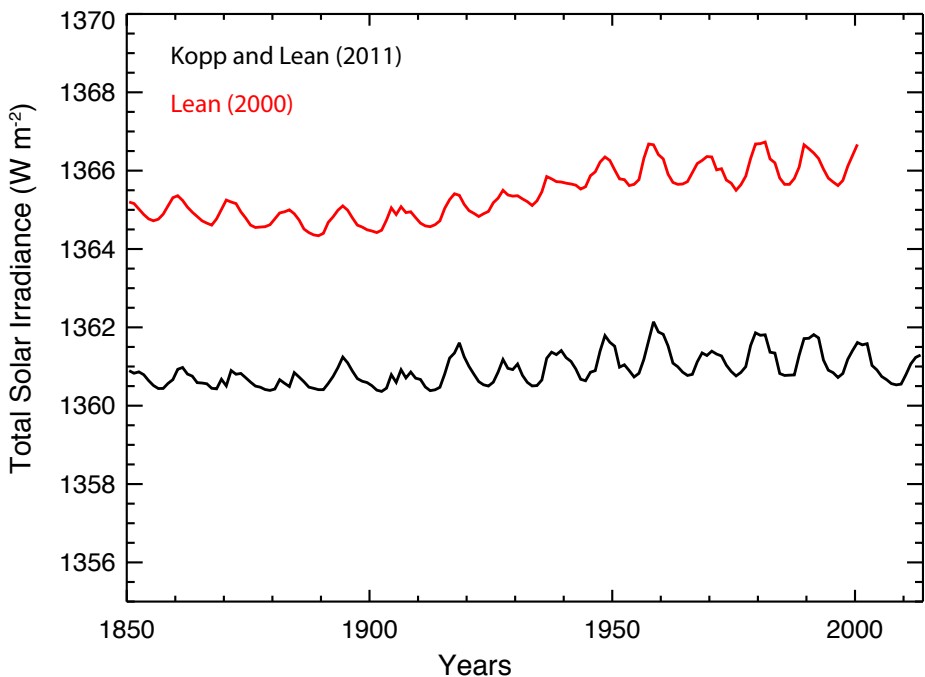

**Figure 2.** Annual mean total solar irradiance. The bias between the Lean (2000) and Kopp and Lean (2011) datasets leads to a reduction in radiative forcing in the new forcing suite.

To identify changes in the forcing time series used to drive the model, we compare the input forcings for the two components for which we have changed datasets. When comparing the forcing time series, only differences in the changes relative to 1860 impact the historical simulations. Time invariant differences are accounted for in the offline Q-flux and initial condition calculations, but differences in the changes are not. In Fig. 2, we show the old and new solar forcing time series. We see that the biggest difference observed in the solar irradiance time series is a bias towards lower values when using the Kopp and Lean (2011) data. The bias is relatively constant at approximately $4.5~\mathrm{Wm}^{-2}$ until 1920, but then increases towards $5.0~\mathrm{Wm}^{-2}$ moving forward in time. The growth of this low bias introduces a weakening of the solar forcing beginning in 1920 in the new suite of forcings.

We observe that ozone concentrations estimated from the AC&C/SPARC dataset differ in both space and time when compared to the previous concentrations used with IGSM (Fig. 3). One clear difference is that the AC&C/SPARC dataset introduces more temporal variability in stratospheric ozone concentrations (which we approximate as pressure levels above 200 mb) prior to 1950. Post-1950, AC&C/SPARC tends to have lower ozone concentrations in the stratosphere and slightly greater concentrations in the troposphere (levels below 200 mb). However, similar to with the solar forcing, we are concerned with the temporal change in the forcing imposed by the ozone concentrations, rather than the relative magnitude of the concentrations across datasets. Beginning in 1900, tropospheric ozone concentrations increase less rapidly in the AC&C/SPARC dataset when

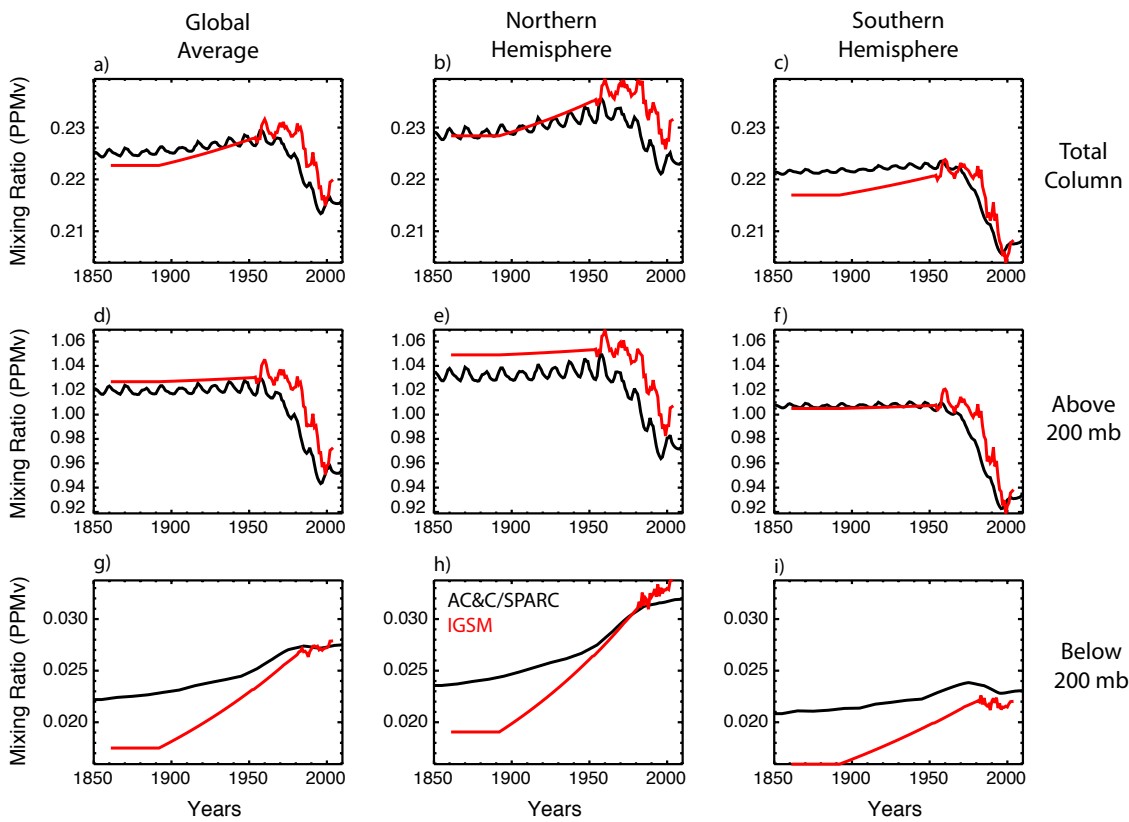

**Figure 3.** Ozone concentration in the old IGSM time series (red) and the Cionni et al. (2011) AC&C/SPARC concentrations (black). (a-c) Annual mean ozone mixing ratio in the total column in the global average (a), northern hemisphere (b), southern hemisphere (c). (d-f) As in (a-c) but for the average above 200 mb. (g-i) As in (a-c) but for the average below 200 mb.

compared to the IGSM dataset. Differences in stratospheric ozone concentrations remain relatively constant until 1950, but then decrease at a slower rate in the AC&C/SPARC time series. These patterns are generally consistent in the global and hemispheric means. When considered separately, increased tropospheric ozone concentrations tend to increase radiative forcing (Stevenson et al., 2013) and decreased stratospheric concentrations tend to increase radiative forcing (Conley et al., 2013). Thus, the less rapid increase in tropospheric ozone concentration and less rapid decrease in stratospheric ozone concentration in the AC&C/SPARC dataset both contribute to a weaker radiative forcing over the historical period in the new suite of forcings.

With the input forcings documented, we focus on deriving probability distributions for the model parameters. We first test the impact of omitting the upper-air diagnostic. As noted in Sect. 3, the surface and upper-air temperature diagnostics are highly correlated. As a result, they reject similar regions of the parameter space for being inconsistent with the observed climate record. Thus, those regions are rejected twice, while regions inconsistent with the ocean heat content diagnostic are rejected

only once. Multiplying the Bayesian likelihood estimate by the same pattern twice leads to a potential bias in the distributions towards regions that are consistent with the surface temperature diagnostic.

Starting from the distributions calculated in Libardoni and Forest (2011), we derive new distributions based only on the surface temperature and ocean heat content diagnostics presented in Sect. 3. We show that reducing the number of diagnostics
from three to two leads to slight changes in the parameter estimates (Table 1). We only present comparisons for ECS and $F_{aer}$ because distributions of $K_v$ were poorly constrained in Libardoni and Forest (2011) and no uncertainty bounds were given. In general, ECS estimates tend to be slightly lower when using only two diagnostics and aerosol estimates are nearly unchanged. Further, the relationships between the distributions with respect to surface dataset are unchanged. Because the changes using only two diagnostics do not change any conclusions from the original study and conservatively removes the risk of double
counting the surface signal, we justify the removal of the upper-air diagnostic.

We next evaluate the impacts that changing the model from IGSM to MESM and updating the forcing suite have on the parameter distributions. We present the new marginal distributions for each parameter in Fig. 4 and observe significant differences between those derived using IGSM and those derived using MESM with the updated forcings (Table 2). Across all datasets, climate sensitivity distributions shift towards higher values and the uncertainty bounds encompass a wider range.
When considering the 90-percent confidence intervals across the distributions derived from each surface dataset, we find climate sensitivity now lies between 1.3 and 5.7 °C, as opposed to the estimated interval of 1.2 to 5.3 °C from Libardoni and Forest (2011). While the uncertainty bounds are still wide compared to other parameters, we observe that $K_v$ is now better constrained with MESM. The distributions of $K_v$ derived using the GISTEMP datasets are still unconstrained with upper tails extending to the edge of the parameter domain, but all other datasets now show an upper bound well within the ensemble range.
We also observe a marked shift in the aerosol estimates. When MESM is used with the updated forcing suite, there is a sizable shift towards weaker aerosol forcing across all datasets. Whereas past estimates put the net aerosol forcing between -0.83 and -0.19 $\mathrm{Wm}^{-2}$, our new estimate of aerosol forcing is between -0.53 and -0.03 $\mathrm{Wm}^{-2}$.

To test whether the differences observed in the parameter estimates were due to the model update, rather than the increased density of model runs, we subsampled each ensemble at the 480 $\theta$s where they overlap (see Fig. 1). We summarize these
distributions in Table 2 and see that there is very little sensitivity when the ensembles are subsampled. Across all datasets, the distributions we derive using the full 640-member IGSM ensemble and those we derive using the 480-member IGSM ensemble are nearly identical for all three parameters. The same is true for the MESM ensemble, except for the distributions we derive for $K_v$. We consistently estimate a smaller upper bound for $K_v$ in the subsampled MESM ensemble compared to when the full MESM ensemble is used. This arises because we assign a probability of zero to regions of the parameter space that have
not been sampled. Thus, for the subsampled MESM ensemble, we assign a probability of zero for $\sqrt{K_v}$ between 5 and 8 cm s$^{-1/2}$, but the likelihood function does not evaluate to zero in this region when using information from the full ensemble. As a result, the full ensemble does not artificially cut off the distribution at $\sqrt{K_v}$ equal to 5 cm s$^{-1/2}$ and leads to higher upper bounds on the distributions. Knowing this, we can conclude from the similarity between distributions derived from the full and subsampled ensembles that the differences we observe between the IGSM and MESM ensembles are due to the differences
between the model and forcing themselves, not the increased density of model runs.

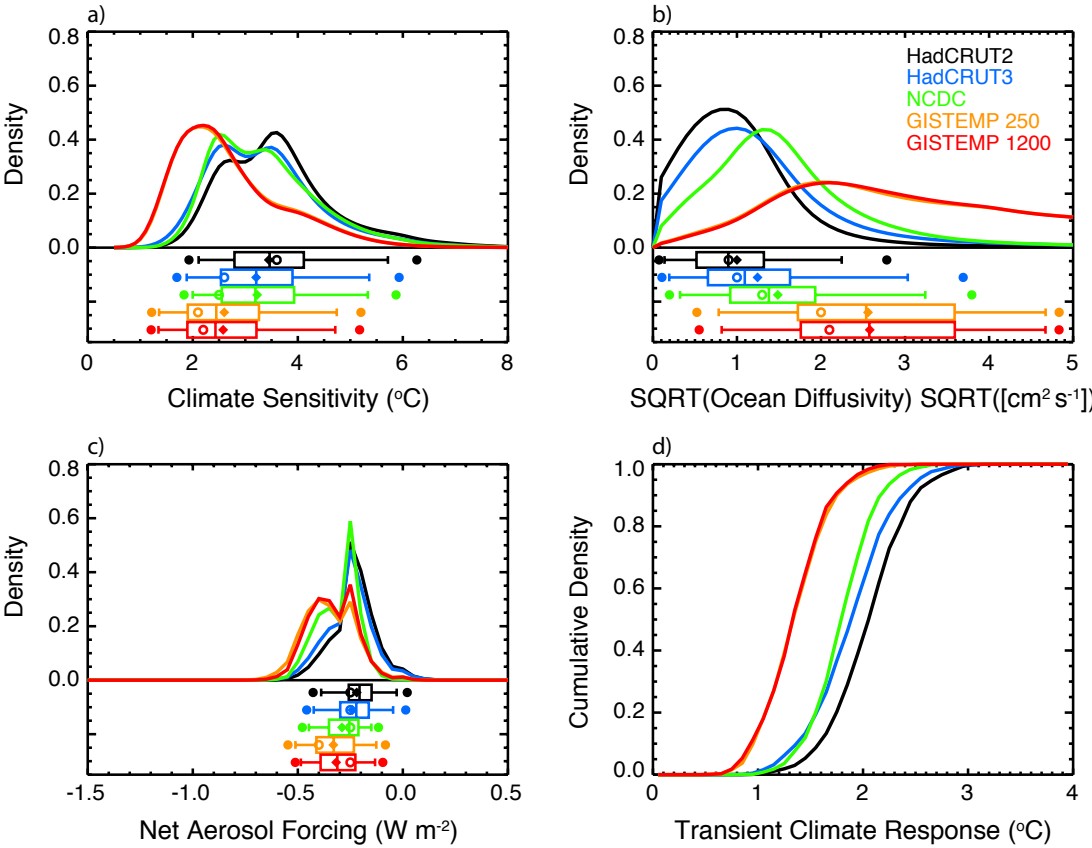

**Figure 4.** Marginal probability distribution functions and TCR cumulative distribution functions derived from MESM simulations using the HadCRUT2, HadCRUT3, NCDC, GISTEMP 250, and GISTEMP 1200 surface temperature datasets as observations: (a) ECS, (b) $K_v$, (c) $F_{aer}$. Whisker plots indicate boundaries for the 2.5-97.5 (dots), 5-95 (vertical bar ends), 25-75 (box ends), and 50 (vertical bar in box) percentiles. Distribution means are represented by diamonds and modes are represented by open circles. (d) TCR CDFs derived from 1000 member Latin Hypercube samples drawn from the joint parameter distributions and the TCR functional fit.

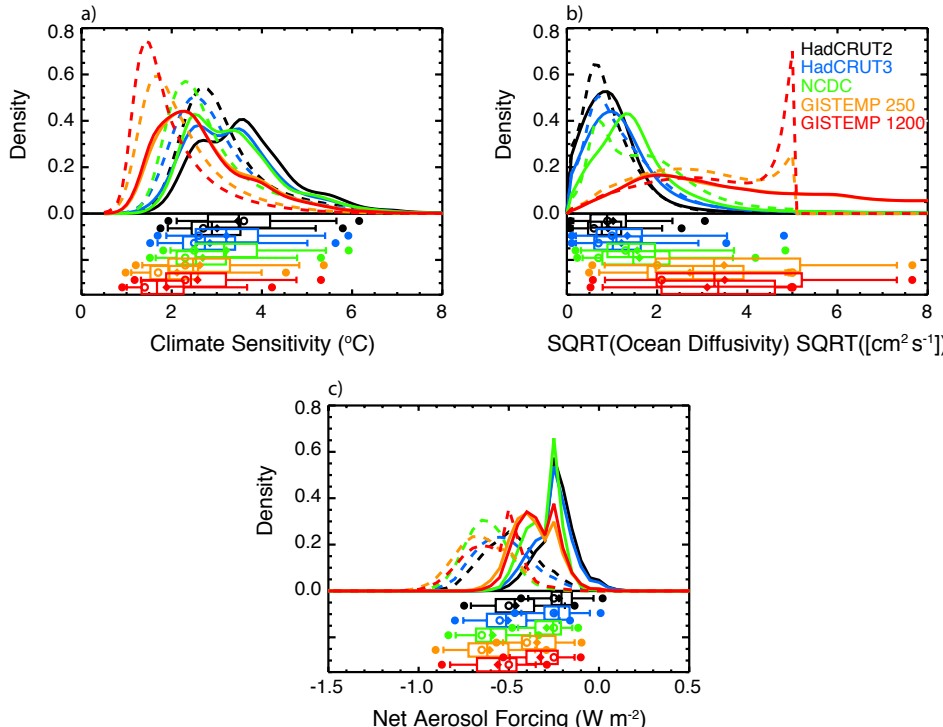

**Figure 5.** Marginal probability distribution functions derived from the full IGSM (dashed) and MESM (solid) ensembles using the Had-CRUT2, HadCRUT3, NCDC, GISTEMP 250, and GISTEMP 1200 surface temperature datasets as observations: (a) ECS, (b) $K_v$, (c) $F_{aer}$. Whisker plots indicate boundaries for the 2.5-97.5 (dots), 5-95 (vertical bar ends), 25-75 (box ends), and 50 (vertical bar in box) percentiles. Distribution means are represented by diamonds and modes are represented by open circles. For a given dataset, the top and bottom whisker plots correspond to the MESM and IGSM ensembles, respectively.

To further demonstrate the total effect of changes to the model, forcings, and ensemble design, we compare the marginal distributions derived from the full IGSM and MESM ensembles using each surface temperature dataset (Fig. 5). For all five datasets, we observe shifts towards higher climate sensitivity, slightly higher ocean diffusivity, and weaker aerosol forcing, consistent with our previous discussion. Further, we demonstrate that the higher ocean diffusivities using the MESM ensembles

5  are the result of not assigning zero probability for $\sqrt{K_v}$ between 5 and 8 $\mathrm{cm\,s^{-1/2}}$. This is clearly evident in the distributions derived using the GISTEMP datasets (Fig. 5b), where the IGSM distributions drop sharply to 0 at $\sqrt{K_v}$ equal to 5 $\mathrm{cm\,s^{-1/2}}$.

Because the parameters are estimated jointly, identifying the causes for specific changes in the marginal distributions are not always straightforward. With this caveat, we now present reasons for the observed changes in the parameter distributions. We begin with $F_{aer}$. As discussed earlier in this section, changes to both the solar and ozone forcing lead to a reduction in

10  their contribution to the global radiation budget. Additionally, there has been a weakening of non-$CO_2$ greenhouse gas forcing introduced by the new radiation code in MESM. These factors result in a decrease in the net radiative forcing on the planet. With the surface temperature and ocean heat content diagnostics unchanged, the same temperature patterns need to be matched

despite the weaker net forcing. One adjustment to the climate system that can help accomplish the matching is to increase the forcing from another term in the energy budget. Of the three model parameters, $F_{aer}$ is the only one that directly changes the radiative forcing, and we thus observe the shift towards less negative aerosol forcing.

An explanation similar to that used for the aerosol distribution can be applied to explaining the observed shifts in the climate sensitivity distribution. In its most basic sense, climate sensitivity is a temperature change per unit forcing. When holding the temperature patterns fixed, the change in temperature is a constant. When explaining the aerosol distribution above, we implicitly fixed the climate sensitivity, requiring the aerosol forcing to be less negative to keep the net forcing constant. However, if we fix $F_{aer}$, the same temperature change needs to be realized with the weaker forcing due to the changes in the solar and ozone forcings. This implies a higher climate sensitivity is required and explains the shifts we observe in the ECS marginal distribution.

In practice, the model parameters are not independent of each other and can change simultaneously. Many combinations of higher climate sensitivity and weaker aerosol forcing lead to similar agreement with the observed temperature record. This suggests a correlation between these two parameters and highlights a strength of estimating the joint PDF for the model parameters: the identification of relationships between the model parameters. However, these relationships also highlight the challenge in attributing changes in a single parameter to a specific cause.

Unlike the climate sensitivity and aerosol forcing distributions, a clear physical explanation for the observed changes in the $K_v$ distribution is more difficult to identify. One reason for this difficulty is the relative insensitivity of the $K_v$ distribution to the model updates. This suggests that either the ocean response is insensitive to changes in the model forcings or that the diagnostics used in this study are unable to constrain the parameter. The latter is explored in a separate study by the authors (Libardoni et al., 2018).

To evaluate how well the model captures the observed record and demonstrate the wide range of climate states simulated by the MESM ensemble, we compare the model output to the observed climate record (Figures 6 and 7). In Fig. 6, we show the global mean surface temperature time series for all ensemble members, along with each of the time series from each of the five observational datasets used in the surface diagnostic. In Fig. 7, we compare the linear trend in the 0-3 km global mean ocean heat content estimated from the MESM simulations against the observed estimate. For both the surface and ocean comparisons, we highlight the estimates from the MESM ensemble members which have parameter settings closest to the median values from the full ensemble MESM distributions.

For both the surface temperature and ocean heat content trends, we have sampled many climate states on the colder and warmer sides of the observed values. We note here that the negative ocean heat content trends are the result of simulations with strong cooling that lie well outside the acceptable range of the parameter space. All simulations with this negative trend have $F_{aer}$ less than or equal to -0.75 $\mathrm{Wm}^{-2}$, a zero-probability region in the MESM ensemble. For the global mean surface temperature time series, the median simulations compare favorably to the observed time series. For the ocean heat content trend, the median simulations tend to overestimate the trend compared to the observed value. Perfect matches should not be expected when comparing the median simulations to the observations, however. Because we derived the distributions using the surface and ocean records, only those runs that agree with both diagnostics are not rejected for being inconsistent with

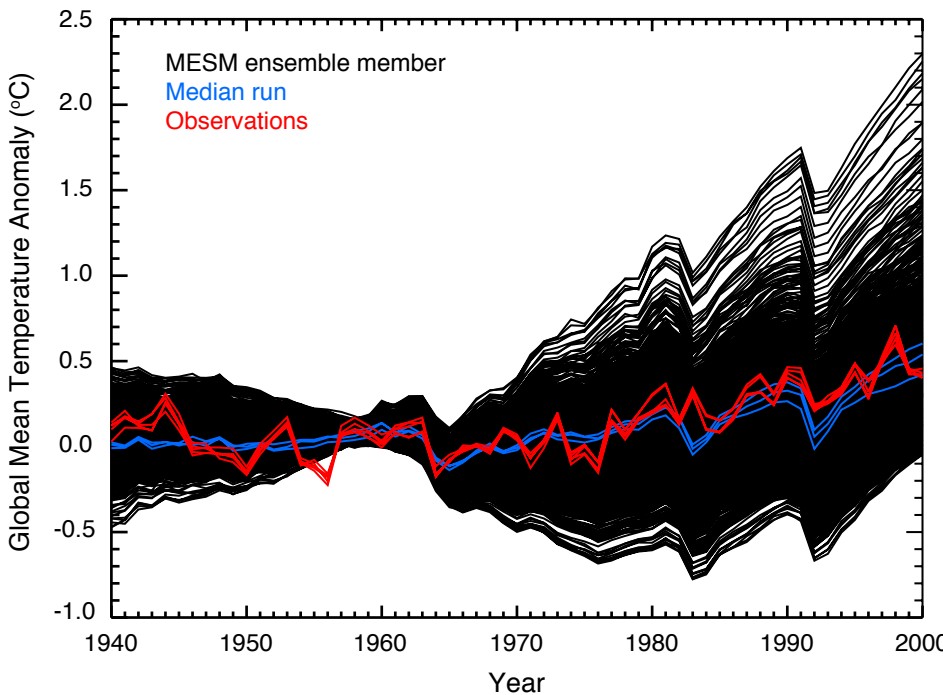

**Figure 6.** Observed and simulated global mean surface temperature anomalies. The observed time series (red) are derived from each of the five surface temperature datasets used in the surface temperature diagnostic. Also shown are the time series for each MESM simulation (black). Runs with parameter settings closest to the median values from each distribution are highlighted (blue). All anomalies are calculated with respect to the 1906-1995 climatology used in the surface diagnostic.

the data. Thus, a model simulation that reproduces the global mean surface temperature perfectly may have too little warming in the deep ocean. Similarly, a model with the perfect ocean heat content trend may not match the surface temperature time series. Small deficiencies in the median runs compared to a single observed record are the result of simultaneously matching the surface and ocean records.

5    To estimate TCR in MESM, we run a 372-member ensemble where all forcings are held fixed and carbon dioxide concentrations are increased by 1% per year. We calculate TCR by estimating the global mean temperature change from the beginning of the simulation to the time of $CO_2$ doubling. Concentrations double in year 70 and we estimate TCR as the average global mean temperature change in years 60 – 80 of the simulation. Temperature changes are calculated with respect to a control simulation with the same model parameters and all forcings held fixed. In a similar manner, we also estimate thermosteric sea

10    level rise (SLR) at the time of doubling. Because all forcings except those attributed to $CO_2$ are fixed, each ECS-$\sqrt{K_v}$ pair yields a single TCR value and a single SLR value, independent of $F_{aer}$.

    We fit a third-order polynomial in ECS and $\sqrt{K_v}$ to the TCR and SLR values calculated from each run to derive a functional fit for all parameter pairs within the domain. The third-order polynomial fit is chosen to be of the same form as the fits derived

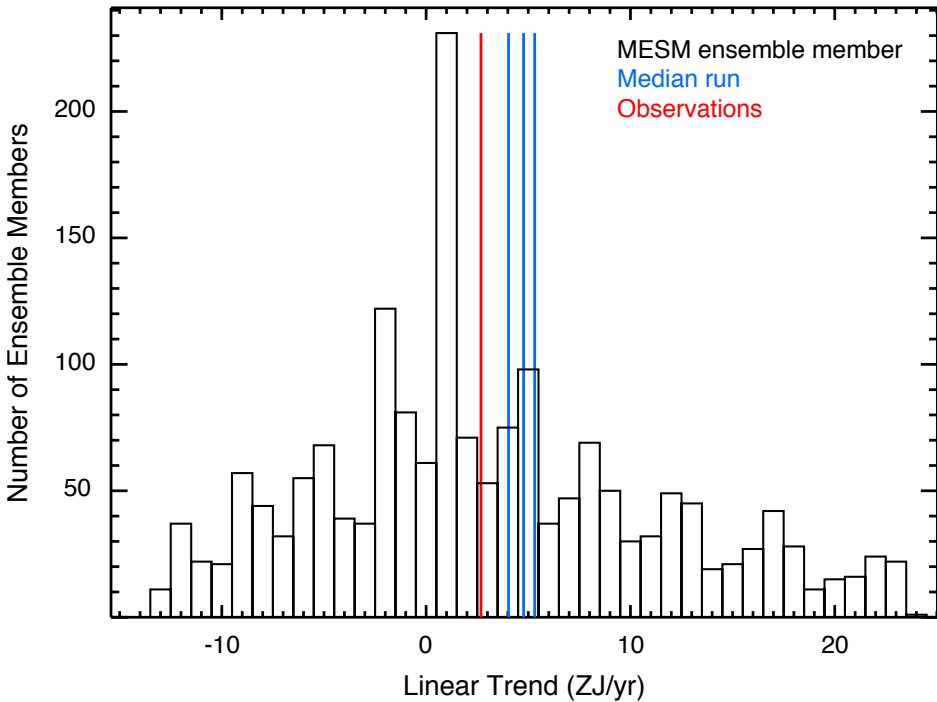

**Figure 7.** Histogram of linear trends in the 0-3 km global mean ocean heat content estimated from each MESM ensemble member. The observed trend (red) and trends estimated from the MESM simulations with parameter values closest to the medians from each distribution (blue) are shown as vertical lines.

for the IGSM model. Further, an investigation of different order fits (not shown) indicated that at least a third-order fit is required to satisfactorily fit the data. From the functional fits, we derive response surfaces for each of the transient properties (Fig. 8). For comparison, we also show the fit derived using the IGSM and its corresponding 1% per year runs, in addition to the differences between the two. Outside of the region where ECS is greater than $4\,°C$ and $\sqrt{K_v}$ is less than about $0.5\,\mathrm{cm\,s^{-1/2}}$
5 and away from the edges of the domain, TCR values from IGSM and MESM agree quite well. There is a similar pattern of agreement in the SLR response surface, with the biggest discrepancies occurring in the high ECS-high $\sqrt{K_v}$ region and near the edges of the parameter domain.

We use the response surface to derive probability distributions for TCR. From each of the joint probability distributions derived using the subsampled MESM ensemble, we draw a 1000-member Latin Hypercube Sample (McKay et al., 1979) of
10 model parameters. The subsampled distributions are chosen so that we restrict the domain to that of the IGSM ensemble, allowing for a more direct comparison of the distributions. Otherwise, high $\sqrt{K_v}$ values that are within the domain of the functional fit to the MESM runs would be selected, for which there is no fit using the IGSM function. We map each of the ECS-$\sqrt{K_v}$ pairs onto the response surface to provide an estimate of TCR values. Binning the responses in a histogram with bin size $= 0.1\,°C$ allows a PDF to be calculated, and the resulting cumulative density functions derived using MESM are displayed

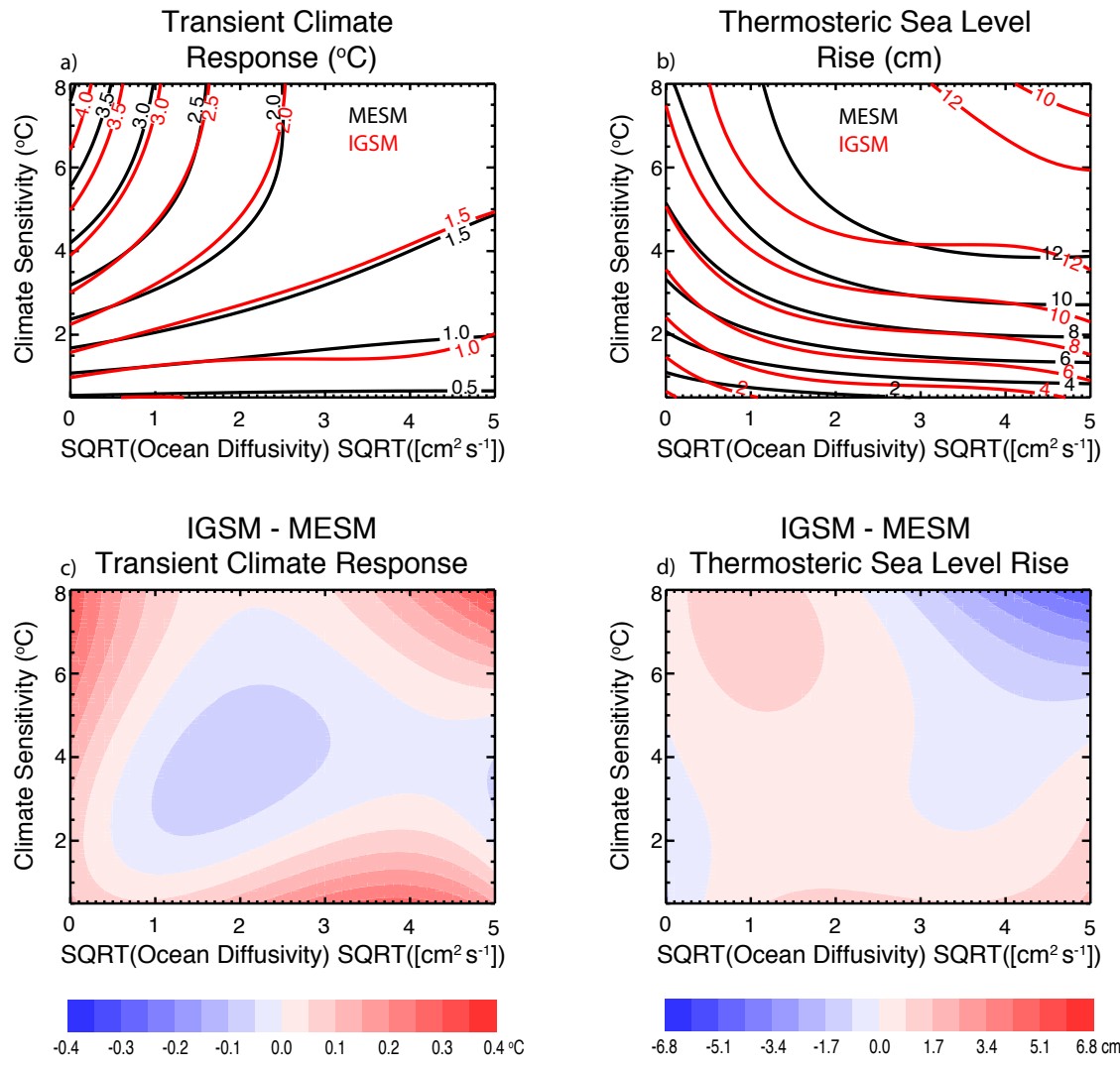

**Figure 8.** Model response surfaces for (a) TCR and (b) thermosteric sea level rise. Contours for the MESM response surfaces are shown in black and contours for the IGSM surfaces are shown in red. Differences between the fits are also shown (c and d).

in Fig. 4d. Comparing TCR distributions for the IGSM and MESM ensembles shows a shift towards higher TCR with the latest results. When comparing the range of 90-percent confidence intervals derived using MESM to those from Libardoni and Forest (2011), we find that TCR estimates increase from 0.87 – 2.31 °C using IGSM to 0.90 – 2.72 °C using MESM. We have shown previously that the marginal distributions of $\sqrt{K_v}$ are similar between the two models, indicating that this shift towards higher
TCR is driven by the higher ECS estimates derived from MESM.

## 5   Conclusions

In this study, we have provided an open, transparent means of testing the changes in model response and parameter estimation to changes in the MIT Integrated Global Systems Model framework. Not only does this systematic accounting of the impacts give a reference point moving forward for studies involving MESM, it proposes a template for assessing the impact that changes
in other simplified climate models have on the calibration of their own model parameters. We hope that this study motivates other modeling groups to perform similar investigations that provide documented accountings of model updates, leading to a more robust understanding of the impacts that the changes have on parameter estimation and model behavior.

By updating the model and its input forcings, we identify the impact that the switch from the MIT Integrated Global Systems Model to the MIT Earth System Model has on the probability distributions of model parameters. The decreases in radiative
forcing due to the change in radiative forcing code, the new solar radiation data, and the new ozone concentrations used to estimate the ozone forcing lead to a net energy deficit when compared to the replaced forcings. This drives an upward shift in our estimates of the 90-percent confidence interval for climate sensitivity from 1.2 to 5.3 °C to 1.3 and 5.7 °C, a better constraint on ocean diffusivity, and a decrease in the 90-percent confidence interval for the net anthropogenic aerosol forcing from between -0.83 and -0.19 $\mathrm{Wm}^{-2}$ to between -0.53 and -0.03 $\mathrm{Wm}^{-2}$. One caveat of our analysis is that because we changed
the forcings and CLM simultaneously, we cannot fully attribute the parameter shifts to the model forcings alone. We have thus shown the total effect of changing both the model and forcings on the parameter distributions, not the effects of the changes individually.

Because TCR is independent of the input forcings, the only difference between the IGSM and MESM configurations in the transient simulations is the land surface model. By showing that the transient climate response surfaces derived from the two
models differ only slightly, we provide evidence that the switch to CLM3.5 does not greatly impact the temperature evolution in the model. We have drawn Latin Hypercube samples from the parameter distributions to provide estimates of TCR from the new response surface. Due to the shift towards higher climate sensitivity and slightly weaker ocean diffusivity, we observe an increase in our 90-percent confidence interval of transient climate response from 0.87 – 2.31 °C to 0.85 – 2.73 °C. By investigating the impact that the new forcings and a newer version of CLM have on the estimates of model parameters and
TCR, we have shown the inherent differences that are present when comparing distributions derived using IGSM and those derived from MESM.

*Code and data availability.* The source code of MESM will become publicly available for non-commercial research and educational purposes as soon as a software license that is being prepared by the MIT Technology Licensing Office is complete. For further information contact mesm-request@mit.edu. A working paper describing and evaluating the MESM is available at http://svante.mit.edu/ mesm/publications/MESM-paper.pdf. All data required to reproduce the figures and tables in the main text and scripts to replicate the figures are available in an online

5   archive. Model output is available upon request.

*Author contributions.* AGL and APS carried out the MESM simulations. APS wrote the codes for extracting model output. AGL performed the analysis and prepared the original manuscript. AGL and CEF developed the model ensemble and experimental design. AGL, CEF, APS, and EM all contributed to interpreting the analysis and synthesizing the findings.

*Competing interests.* The authors declare no competing interests.

10   *Acknowledgements.* This work was supported by U.S. Department of Energy (DOE), Office of Science under award DE-FG02-94ER61937 and other government, industry and foundation sponsors of the MIT Joint Program on the Science and Policy of Global Change. For a complete list of sponsors and U.S. government funding sources, see http://globalchange.mit.edu/sponsors/. The authors would like to thank the National Climatic Data Center, the Hadley Centre for Climate Prediction and Research, and the NASA Goddard Institute for Space Studies for producing publicly available surface data products and the NOAA National Centers for Environmental Information for providing publicly

15   available ocean heat content data. We would also like to thank the University of Maryland for access to the Evergreen high-performance computing cluster for model simulations.

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

**Table 1.** 90-percent confidence intervals for climate sensitivity (ECS) and net aerosol forcing ($F_{aer}$). Distributions that include the upper-air diagnostic are from Libardoni and Forest (2011) and distributions with two diagnostics exclude the upper-air diagnostic.

| Surface Temperature Dataset | # Diagnostics | ECS (°C) | | $F_{aer}$ (Wm$^{-2}$) | |
|---|---|---|---|---|---|
| | | 5% | 95% | 5% | 95% |
| HadCRUT2[1] | 3 | 2.0 | 5.3 | -0.19 | -0.70 |
| | 2 | 1.9 | 5.2 | -0.19 | -0.71 |
| HadCRUT3[2] | 3 | 1.9 | 5.1 | -0.22 | -0.74 |
| | 2 | 1.7 | 5.0 | -0.38 | -0.79 |
| NCDC[3] | 3 | 1.8 | 4.7 | -0.37 | -0.78 |
| | 2 | 1.6 | 4.8 | -0.38 | -0.79 |
| GISTEMP250[4] | 3 | 1.3 | 3.6 | -0.32 | -0.83 |
| | 2 | 1.1 | 4.0 | -0.35 | -0.83 |
| GISTEMP1200[5] | 3 | 1.2 | 3.4 | -0.33 | -0.80 |
| | 2 | 1.0 | 3.7 | -0.35 | -0.83 |

[1]Hadley Centre Climatic Research Unit Temperature version 2 (Jones and Moberg, 2003)

[2]Hadley Centre Climatic Research Unit Temperature version 3 (Brohan et al., 2006)

[3]National Climatic Data Center merged land-ocean dataset (Smith et al., 2008)

[4]GISS Surface Temperature Analysis with 250 km smoothing (Hansen et al., 2010)

[5]GISS Surface Temperature Analysis with 1200 km smoothing (Hansen et al., 2010)

**Table 2.** 90-percent confidence intervals and means for climate sensitivity (ECS), ocean diffusivity ($K_v$), and net aerosol forcing ($F_{aer}$). Surface temperature datasets are the same as in Table 1.

| Surface Temperature Dataset | Model and Runs | ECS ($^\circ$C) | | | $\sqrt{K_v}$ (cm s$^{-1/2}$) | | | $F_{aer}$ (Wm$^{-2}$) | | |
|---|---|---|---|---|---|---|---|---|---|---|
| | | 5% | 95% | Mean | 5% | 95% | Mean | 5% | 95% | Mean |
| HadCRUT2 | Full IGSM | 1.9 | 5.2 | 3.0 | 0.1 | 2.1 | 0.9 | -0.19 | -0.71 | -0.46 |
| | Subsampled IGSM | 1.9 | 5.2 | 3.0 | 0.1 | 2.1 | 0.9 | -0.16 | -0.71 | -0.45 |
| | Full MESM | 2.1 | 5.7 | 3.5 | 0.1 | 2.3 | 1.0 | -0.03 | -0.39 | -0.22 |
| | Subsampled MESM | 2.1 | 5.7 | 3.4 | 0.1 | 2.2 | 1.0 | -0.03 | -0.39 | -0.22 |
| HadCRUT3 | Full IGSM | 1.7 | 4.0 | 2.8 | 0.2 | 2.9 | 1.2 | -0.22 | -0.75 | -0.50 |
| | Subsampled IGSM | 1.7 | 4.0 | 2.8 | 0.2 | 2.9 | 1.2 | -0.20 | -0.75 | -0.49 |
| | Full MESM | 1.9 | 5.4 | 3.2 | 0.2 | 3.6 | 1.3 | -0.05 | -0.43 | -0.24 |
| | Subsampled MESM | 1.9 | 5.4 | 3.2 | 0.2 | 3.0 | 1.2 | -0.05 | -0.42 | -0.24 |
| NCDC | Full IGSM | 1.6 | 4.8 | 2.7 | 0.3 | 3.7 | 1.6 | -0.38 | -0.79 | -0.59 |
| | Subsampled IGSM | 1.6 | 4.8 | 2.7 | 0.3 | 3.7 | 1.6 | -0.36 | -0.79 | -0.58 |
| | Full MESM | 2.0 | 5.4 | 3.2 | 0.3 | 3.7 | 1.6 | -0.15 | -0.45 | -0.29 |
| | Subsampled MESM | 2.0 | 5.3 | 3.2 | 0.3 | 3.2 | 1.5 | -0.15 | -0.45 | -0.29 |
| GISTEMP 250 | Full IGSM | 1.1 | 4.0 | 2.1 | 0.7 | 4.8 | 2.7 | -0.35 | -0.86 | -0.61 |
| | Subsampled IGSM | 1.1 | 4.0 | 2.1 | 0.6 | 4.8 | 2.7 | -0.35 | -0.86 | -0.60 |
| | Full MESM | 1.3 | 4.8 | 2.6 | 0.8 | 7.3 | 3.5 | -0.13 | -0.53 | -0.34 |
| | Subsampled MESM | 1.4 | 4.7 | 2.6 | 0.8 | 4.7 | 2.6 | -0.13 | -0.51 | -0.33 |
| GISTEMP 1200 | Full IGSM | 1.0 | 3.7 | 1.9 | 0.8 | 4.9 | 3.1 | -0.35 | -0.83 | -0.56 |
| | Subsampled IGSM | 1.0 | 3.7 | 1.9 | 0.7 | 4.9 | 3.1 | -0.35 | -0.82 | -0.56 |
| | Full MESM | 1.3 | 4.8 | 2.6 | 0.8 | 7.3 | 3.5 | -0.14 | -0.49 | -0.33 |
| | Subsampled MESM | 1.3 | 4.7 | 2.6 | 0.8 | 4.7 | 2.6 | -0.14 | -0.49 | -0.32 |