# Peer review of "Baseline Evaluation of the Impact of Updates to the MIT Earth System Model on its Model Parameter Estimates"

_Geoscientific Model Development, 2018_

## Referee Comment (RC1) · Anonymous Referee #1 · 1 May 2018

Review of:

Baseline evaluation of the impact of updates to the MIT Earth System Model on its model parameter estimates

By:

A. G. Libardoni, C. E. Forest, A. P. Sokolov, and E. Monier

Submitted to: Geoscientific Model Development

General Comments:

This is a well-written and interesting paper documenting new updates to the MIT Earth System Model. In particular, the authors highlight how updated forcings and changes to the land surface model influence model parameter estimates of equilibrium climate sensitivity and the transient climate response (TCR).

This is a useful paper that is well-suited for GMD. However, it is a bit brief and lacks sufficient depth for GMD. In particular, I think the paper falls a bit short in providing enough details about the model "development", which is key for this journal. I recommend adding more description of the model and key changes since the last version (right now the descriptions of the model and forcings are limited to two dense paragraphs), or perhaps a simple summary of the model lineage and history and/or recent results from the authors' companion papers? Additionally, it would be nice to see more explanation about what causes the differences between model versions and connections with broader climate problems (e.g. probabilistic projections) beyond the reporting of the model sensitivities... such as more description of the importance (or lack thereof) of ocean heat uptake and aerosol forcing, and how new model parameter estimates compare to previous versions. I provide additional specific comments and suggestions below:

Specific Comments:

P1, L10: "absent an increase" is odd wording. P1, L12: This statemetn seems redundant to line 4-5 P1, L14-15: What causes these shifts? P1, L15-16: So if the land surface model has limited effect on temperature evolution, is it updates to the forcings that cause the differences in climate sensitivity estimates? It's not entirely clear what points the authors are trying to convey here. I suggest tightening up the abstract to highlight the significance.

P7, last paragraph: The authors raise interesting, but somewhat contradictory, points. They state that reducing the number of diagnostics from 3 to 2 has little impact on model parameter estimates, but then go on to state that CS estimates are lower when
using 2 diagnostics. Why are the results insensitive to the upper-air diagnostic? Also, the constraint on Kv is not clear. Is there any update since what was shown their previous work (e.g. Libardoni and Forest 2011)? I suggest adding more details to these points to help the reader.

P8, L10: Can you show a plot of the ECS pdf for IGSM and MESM for comparison?

P9, L3-4: How do these new estimates of net aerosol forcing compare with other recent estimates?

P10: L14: I'm a little unclear how ocean diffusivity fits in with the analysis. Why did the old ensemble cut of high values of Kv? It is also relatively insensitive to the model updates compared to aerosol forcing and equilibrium climate sensitivity. Why is this? I recommend the authors streamline the results and discussion sections to include a summary of key points about each model parameter, the constraints and model sensitivities, and physical reasoning for the differences.

P12, L7: Why choose a third-order polynomial here? Is there sensitivity in the fits to the functional form? Would you expect similar results in terms of model differences using a 2nd order polynomial?

P12, L24-25: The authors state that the shift towards higher transient climate response is driven by higher climate sensitivity in MESM, but there is not enough explanation in my opinion as to why there is a larger CS in MESM compared to previous versions, how they compare (e.g. posterior distributions), and to what extent the updated forcings play a role.

P12-13: The conclusions provide a nice summary of the paper's key points. I suggest expanding the results section to include more in-depth discussion along these lines.

---

## Referee Comment (RC2) · Anonymous Referee #2 · 14 May 2018

The authors compare the most recent version of the MIT Earth System Model (MESM) with an earlier version and document the impact of updating changes in external forcing over the historical period and of the land surface scheme. They present results from an 1800-member ensemble simulation over the industrial period and compare the model results to two observational metrics of change (ocean heat uptake, surface air temperature pattern). They also present a 372-member ensemble in idealized forcing scenarios to establish the links between TCR, thermosteric sea level rise, ECS, and ocean heat uptake. The illustration of these links in Figure 5 is nice and interesting. On the less positive side, it is surprising that the authors do not invoke a larger suite of observational constraints to estimate probability density distributions of TCR and other

model outcomes. The method section requires more work and there is also a lack of discussion in section 5.

1) I miss a description of the basic components and parameterizations of the model in the method section 3.

I miss a section that describes model spin-up and the setup for the different model simulations, including external forcing factors.

Further, it is not evident from the description why the model is called "Earth System Model". For example, are biogeochemical cycles included? Does dynamic vegetation affect albedo? Is it an ESM or rather an Earth System Model of Intermediate Complexity?

I also miss a brief description of the metric used to compare model and data and how they are used to derive probability distribution. It is not sufficient to refer the reader to the literature (Libardoni and Forest 2011).

2) Section 3: The authors vary three parameters – ocean diffusivity, an aerosol forcing scaling, and the strength of the cloud feedback determining ECS and constrain the models with two parameters.

2a) There is little information in the method section what these parameters specifically influence. The aerosol forcing scaling is unclear. Does this mean that all aerosol forcings are lumped together and scaled with a constant time invariant factor? How are different uncertainties applying to different aerosol classes (e.g. sulfate versus soot) considered or not and what is the justification for this approach. Please discuss caveats related to your assumption of a scaling factor.

2b) Effective ocean diffusivity is a very loose term. Is this diapycnal, vertical or horizontal diffusivity or does the parameter refer to the diffusivity associated with Gent-McWilliams parameterization? The subscript v of $K_v$ points to vertical diffusivity. I would hope that this parameter reflects diapycnal diffusivity as diapycnal diffusivity cogoverns ocean overturning strength and thus surface-to-deep heat transport. In any case, I am puzzled about the range sampled. Diapycnal diffusivity in coarse resolution, dynamic ocean models is typically of order 0.1 10-4 m2 s-1. Here diffusivity is varied in steps of 1 10-4 m2 s-1 and a very wide range up to 64 10-4 m2 s-1 is used. The upper value is even much larger than applied in classical box-diffusion models (1-2 10-4 m2 s-1 ); in box-diffusion models the entire vertical transport (mixing, advection, convection) is parameterized by diffusion only. What is the justification for this large sampling range? As a minor point, please use SI units for diffusivity. Further, I though Gent-McWilliams parameterization is included in the MIT model. If yes, why is the Gent-McWilliams diffusivity not varied or is this parameter linked with the "effective diffusivity"?

2c) ECS is typically used to abbreviate Equilibrium Climate Sensitivity. Here, an effective climate sensitivity is introduced and termed ECS. What represents this effective climate sensitivity?

3) Section 3: I question somewhat the application of only two observational metrics to constrain ECS, TCR, and sea level rise. Namely, pattern of surface air temperature change and "linear" ocean heat uptake are used as constraints by the authors. In my opinion, there is a lack of observational constraints to probe the timescales of deep ocean overturning (e.g. 14C). Thus it appears not surprising that the diffusivity parameter remains not well constrained.

There is also a lack of metrics to probe the spatial pattern of heat uptake. This is particularly important as the thermal expansion coefficient varies by almost an order of magnitude in the ocean. Thus it matters, where the heat is taken up to estimate sea level rise.

As another focus of the study is on TCR, it would also be nice to invoke additional metrics on thermocline ventilation as for example available by observation-derived fields of CFCs and bomb-produced 14C.

4) Page 5 to page 7, results, The description of the difference in input forcing is useful, but in my opinion misplaced. Solar and ozone forcings are model drivers (or forcings) and distinct from a particular model version. These forcings should be described in the method section where the simulations and the applied external forcings are to be described.

5) P6, line 3ff; Q-flux adjustment: Does this mean that the authors apply temperature flux correction to their model? This should be explained in the method section.

6) Section 4: I miss a figure comparing the modelled pattern of the median (or mean or best-guess version) with the observed pattern of surface air temperature change and similar for the global ocean heat uptake and its spatial pattern (and may be for upper air temperature) to illustrate how well the model is able to capture the observations.

7) Page 12, line 7: How well does the polynomial fit represent the model results?

8) Page 12, line 14: Why is the PDF for the TCR not directly estimated from the 372-member ensemble? Does the fitting add additional uncertainties to the procedure of estimating TCR?

9) Discussion and conclusion: While the authors suggest that their approach should serve as a template for other groups, they fail to mention that similar, and sometime much more comprehensive approaches of parameter calibration, have been undertaken by other groups. They also fail to compare their estimate of TCR and ECS with published estimate and to put their findings in the context of the wider literature. See for example, Collins et al., IPCC, 2013 for the most recent assessment of TCR and ECS values by IPCC. Of course there are recent updates of these estimates and there are also many other studies that determine model parameters such as vertical ocean diffusivity. Examples that come immediately in my mind are Holden et al., Clim. Dyn., 2010, Richardson; Nat. Clim.Change, 2016, Schmittner et al., GBC, 2009, Steinacher et al., Science, 2013 or Steinacher and Joos, Biogeosciences 2016. It is the task of the authors to identify the recent literature to provide a relevant discussion.

P1, Line 22: typo: sensitivity
* * *

---

## Author Comment (AC1) · 2 Jul 2018

P1, L10: "absent an increase" is odd wording. P1, L12: This statemetn seems redundant to line 4-5 P1, L14-15: What causes these shifts? P1, L15-16: So if the land surface model has limited effect on temperature evolution, is it updates to the forcings that cause the differences in climate sensitivity estimates? It's not entirely clear what points the authors are trying to convey here. I suggest tightening up the abstract to highlight the significance.

Response: Per these comments, we have revised our abstract to make the summary clearer. We attribute the observed shifts in the parameter distributions to the changes

in model forcings. The land surface model impacts other components of MESM (i.e., carbon fluxes), but in the climate component used here, it has little impact.

P7, last paragraph: The authors raise interesting, but somewhat contradictory, points. They state that reducing the number of diagnostics from 3 to 2 has little impact on model parameter estimates, but then go on to state that CS estimates are lower when using 2 diagnostics. Why are the results insensitive to the upper-air diagnostic? Also, the constraint on Kv is not clear. Is there any update since what was shown their previous work (e.g. Libardoni and Forest 2011)? I suggest adding more details to these points to help the reader.

Response: We have cleared up these points by adding discussions into the manuscript. The main reason for omitting the upper-air diagnostic is the significant correlation between the upper-air temperature pattern and the surface temperature pattern as a result of the lapse rate and water vapor feedbacks. Each of these diagnostics reject similar regions of the parameter space for being inconsistent with the observed climate record, thus potentially double counting the same temperature response signal. Removing the upper-air diagnostic removes the risk of bias due to treating it as a statistically independent diagnostic.

There has not been any additional work on constraining Kv between our previous work and this manuscript. Currently, a second publication is in review (Libardoni et al., 2018, ASCMO) that investigates how including additional data in the model diagnostics improves the model parameter estimates. We show there that including additional data improves the model diagnostics and leads to better constraint on Kv. We chose not to incorporate any changes to the model diagnostics in this manuscript to provide a clean comparison of changes resulting from changing only the model version.

P8, L10: Can you show a plot of the ECS pdf for IGSM and MESM for comparison?

Response: For each of the distributions derived from the individual surface temperature datasets, we plotted the marginal PDFs for the full IGSM and MESM ensembles.

In all five cases, the same changes are observed: higher climate sensitivity, nearly unchanged ocean diffusivity, and weaker negative aerosol forcing.

P9, L3-4: How do these new estimates of net aerosol forcing compare with other recent estimates?

Response: For EMICs like MESM, the net aerosol forcing is a model-specific parameter, making a clear comparison between studies and direct observations challenging. For example, the aerosol forcing pattern may account for different model forcings and be defined for different time periods. For example, while Andronova and Schlesinger (2001) scale the natural and anthropogenic aerosol direct and indirect forcings by adjusting the amplitude in 1990, the aerosol parameter in Knutti et al. (2002) is scaled in 2000 and represents the indirect aerosol effect and any other forcing not explicitly represented in the model. With these differences in mind, estimates of aerosol forcing from energy balance models and EMICs fall in the ranges -1.3 to -0.54 W/mˆ2 (Andronova and Schlesinger, 2001), -1.2 to 0 W/mˆ2 (Knutti et al., 2002), -1.53 to -0.33 W/mˆ2 (Kriegler, 2005), -0.83 to -0.19 W/mˆ2 (Libardoni and Forest, 2011), and -1.7 to -0.4 W/mˆ2 (Skeie et al., 2014).

P10: L14: I'm a little unclear how ocean diffusivity fits in with the analysis. Why did the old ensemble cut of high values of Kv? It is also relatively insensitive to the model updates compared to aerosol forcing and equilibrium climate sensitivity. Why is this? I recommend the authors streamline the results and discussion sections to include a summary of key points about each model parameter, the constraints and model sensitivities, and physical reasoning for the differences.

Response: Kv fits into the analysis because all three model parameters are estimated jointly, with the marginal PDFs calculated by integrating the joint PDF over the other two model parameters. Thus, changes due to the model and forcings can impact any of the three marginal distributions. As we point out in the edited manuscript, physical explanations for the changes in the ECS and aerosol distributions are more accessible

than an explanation for Kv. However, because all three parameters are estimated together, changes in the other two parameters can impact our Kv estimates.

In both ensembles, values of Kv outside the of range of values sampled are assigned zero probability. This meant assigning zero probability for regions greater than 5 cm/sˆ1/2 for the IGSM ensemble and 8 cm/sˆ1/2 for the MESM ensemble. From the full MESM ensemble, we find non-zero, although small, probabilities of Kv between 5 and 8 cm/sˆ1/2. By accounting for the extra mass in the tail regions for the MESM ensembles, the Kv quantiles are pulled towards higher values.

We have added text to the manuscript explaining the points above. Further, we have re-ordered the results section to devote separate paragraphs for each parameter that explains the changes in the marginal distributions separately. This makes the key points stand out more clearly.

P12, L7: Why choose a third-order polynomial here? Is there sensitivity in the fits to the functional form? Would you expect similar results in terms of model differences using a 2nd order polynomial?

Response: The third-order polynomial was chosen for consistency with previous work to provide the most direct comparison possible between the surfaces derived for IGSM and MESM. In offline tests, we derived additional surfaces for first-, second-, and fourth-order polynomial fits and compared them to the TCR and SLR values calculated directly from the transient simulations. The first-order approximation leads to an unsatisfactory fit with gradients of TCR and SLR in the Kv direction that are too weak. The second-order fit produces curvature in TCR and SLR contours that are inconsistent with those calculated directly from the transient simulations. In particular, the 1.5 C contour for TCR using the second-order fit suggested that for a single Kv value, two different ECS values could be used. Further, the second-order fit shows that sea level rise greater than 14 cm is possible within the sampled domain, whereas none of the transient simulations had SRL that high. The third- and fourth-order fits both showed

good agreement with the simulated results but were not without their flaws. The third-order fit showed some error in the 1.5 C TCR contour, where the fourth-order fit led to regions of SLR greater than 14 cm within the domain. We mention these tests in the revised manuscript and keep the third-order fit to maintain consistency with previous work. Improving this fit is a potential avenue of future research.

P12, L24-25: The authors state that the shift towards higher transient climate response is driven by higher climate sensitivity in MESM, but there is not enough explanation in my opinion as to why there is a larger CS in MESM compared to previous versions, how they compare (e.g. posterior distributions), and to what extent the updated forcings play a role.

Response: Through the points made above and the changes made to the manuscript, we believe that this has been addressed more clearly. We have added discussions for each parameter that specifies how the changes to the model forcings could lead to the observed shifts in the marginal distributions. Looking at the response surfaces, for any Kv value, an increase in ECS leads to larger TCR. Thus, given a constant Kv distribution, shifts towards higher ECS result in a shift towards higher TCR. With the relatively small changes in the Kv distribution from the subsampled MESM ensemble (see Table 2 of the manuscript), we find the assumption of constant Kv distribution needed for this argument justifiable.

P12-13: The conclusions provide a nice summary of the paper's key points. I suggest expanding the results section to include more in-depth discussion along these lines.

Response: As noted above, we have expanded the results section to provide more in-depth discussion of the reasons for the changes in the parameter and TCR estimates.

---

## Author Response (AR1)

**Response to Reviewer #1**

*P1, L10: "absent an increase" is odd wording.*
*P1, L12: This statemetn seems redundant to line 4-5 P1, L14-15: What causes these shifts?*
*P1, L15-16: So if the land surface model has limited effect on temperature evolution, is it updates to the forcings that cause the differences in climate sensitivity estimates? It's not entirely clear what points the authors are trying to convey here. I suggest tightening up the abstract to highlight the significance.*

Response:  We attribute the observed shifts in the parameter distributions to the changes in model forcings.  The land surface model impacts other components of MESM (i.e., carbon fluxes), but in the climate component used here, it has little impact.

Changes:  Per these comments, we have revised our abstract to make the summary clearer.  An explicit statement has been added that addresses the reason for changes to the distributions.

*P7, last paragraph: The authors raise interesting, but somewhat contradictory, points. They state that reducing the number of diagnostics from 3 to 2 has little impact on model parameter estimates, but then go on to state that CS estimates are lower when using 2 diagnostics. Why are the results insensitive to the upper-air diagnostic? Also, the constraint on Kv is not clear. Is there any update since what was shown their previous work (e.g. Libardoni and Forest 2011)? I suggest adding more details to these points to help the reader.*

Response:  The main reason for omitting the upper-air diagnostic is the significant correlation between the upper-air temperature pattern and the surface temperature pattern as a result of the lapse rate and water vapor feedbacks.  Each of these diagnostics reject similar regions of the parameter space for being inconsistent with the observed climate record, thus potentially double counting the same temperature response signal.  Removing the upper-air diagnostic removes the risk of bias due to treating it as a statistically independent diagnostic.

There has not been any additional work on constraining Kv between our previous work and this manuscript.  Currently, a second publication is in review (Libardoni et al., 2018, ASCMO) that investigates how including additional data in the model diagnostics improves the model parameter estimates.  We show there that including additional data improves the model diagnostics and leads to better constraint on Kv.  We chose not to incorporate any changes to the model diagnostics in this manuscript to provide a clean comparison of changes resulting from changing only the model version.

Changes:  We have cleared up these points by adding clarifying remarks into the manuscript.  In Section 3, we include a discussion of why multiple diagnostics are preferable and why independent diagnostics of model performance are important.  Further, we provide a reference study that addresses the correlation of the surface and upper-air diagnostics.  This point is highlighted further in Section 4 when discussing the changes in the PDFs resulting from the

reduction in the number of diagnostics. Contradictory language regarding the size and significance of the changes when moving from three to two diagnostics has been removed for clarity.

*P8, L10: Can you show a plot of the ECS pdf for IGSM and MESM for comparison?*

Response: For each of the distributions derived from the individual surface temperature datasets, we plotted the marginal PDFs for the full IGSM and MESM ensembles. In all five cases, the same changes are observed: higher climate sensitivity, nearly unchanged ocean diffusivity, and weaker negative aerosol forcing.

Changes: We have added this figure (Figure 5) and supporting text in Section 4.

*P9, L3-4: How do these new estimates of net aerosol forcing compare with other recent estimates?*

Response: For EMICs like MESM, the net aerosol forcing is a model-specific parameter, making a clear comparison between studies and direct observations challenging. For example, the aerosol forcing pattern may account for different model forcings and be defined for different time periods. For example, while Andronova and Schlesinger (2001) scale the natural and anthropogenic aerosol direct and indirect forcings by adjusting the amplitude in 1990, the aerosol parameter in Knutti et al. (2002) is scaled in 2000 and represents the indirect aerosol effect and any other forcing not explicitly represented in the model. With these differences in mind, estimates of aerosol forcing from energy balance models and EMICs fall in the ranges -1.3 to -0.54 $Wm^{-2}$ (Andronova and Schlesinger, 2001), -1.2 to 0 $Wm^{-2}$ (Knutti et al., 2002), -1.53 to -0.33 $Wm^{-2}$ (Kriegler, 2005), -0.83 to -0.19 $Wm^{-2}$ (Libardoni and Forest, 2011), and -1.7 to -0.4 $Wm^{-2}$ (Skeie et al., 2014).

Changes: No major changes have been made to the manuscript in response to this comment. We have intentionally left the comparison of our parameter estimates with other groups for our other studies. This is done to place the emphasis of this work on setting the baseline for how the change in the forcings and model impact the parameter and TCR estimates.

*P10: L14: I'm a little unclear how ocean diffusivity fits in with the analysis. Why did the old ensemble cut of high values of Kv? It is also relatively insensitive to the model updates compared to aerosol forcing and equilibrium climate sensitivity. Why is this? I recommend the authors streamline the results and discussion sections to include a summary of key points about each model parameter, the constraints and model sensitivities, and physical reasoning for the differences.*

Response: Kv fits into the analysis because all three model parameters are estimated jointly, with the marginal PDFs calculated by integrating the joint PDF over the other two model parameters. Thus, changes due to the model and forcings can impact any of the three marginal distributions. As we point out in the edited manuscript, physical explanations for the changes

in the ECS and aerosol distributions are more accessible than an explanation for Kv. However, because all three parameters are estimated together, changes in the other two parameters can impact our Kv estimates.

In both ensembles, values of Kv outside the of range of values sampled are assigned zero probability. This meant assigning zero probability for regions greater than 5 cms$^{-1/2}$ for the IGSM ensemble and 8 cms$^{-1/2}$ for the MESM ensemble. From the full MESM ensemble, we find non-zero, although small, probabilities of Kv between 5 and 8 cms$^{-1/2}$. By accounting for the extra mass in the tail regions for the MESM ensembles, the Kv quantiles are pulled towards higher values.

Changes: We have added text to the manuscript addressing the points above. Beginning on Page 11, Line 20 of the revised manuscript, we provide discussions of each model parameter as suggested by the reviewer. As part of the discussions, we give physical explanations to support the changes we observe in the parameter distributions. We further strengthen the discussion by including the benefits and challenges of estimating the parameters together. In particular, we highlight that the joint distribution allows us to identify correlations amongst the parameters, but also makes the attributing the changes in a single parameter to one cause less straightforward.

An explicit explanation for the cut-off of high Kv values is given beginning on Page 11, Line 7 of the revised manuscript. The insensitivity of the Kv distribution is addressed in the paragraph devoted to the parameter (Page 12, Line 6).

*P12, L7: Why choose a third-order polynomial here? Is there sensitivity in the fits to the functional form? Would you expect similar results in terms of model differences using a 2nd order polynomial?*

Response: The third-order polynomial was chosen for consistency with previous work to provide the most direct comparison possible between the surfaces derived for IGSM and MESM. In offline tests, we derived additional surfaces for first-, second-, and fourth-order polynomial fits and compared them to the TCR and SLR values calculated directly from the transient simulations. The first-order approximation leads to an unsatisfactory fit with gradients of TCR and SLR in the Kv direction that are too weak. The second-order fit produces curvature in TCR and SLR contours that are inconsistent with those calculated directly from the transient simulations. In particular, the 1.5 °C contour for TCR using the second-order fit suggested that for a single Kv value, two different ECS values could be used. Further, the second-order fit shows that sea level rise greater than 14 cm is possible within the sampled domain, whereas none of the transient simulations had SLR that high. The third- and fourth-order fits both showed good agreement with the simulated results but were not without their flaws. The third-order fit showed some error in the 1.5 °C TCR contour, where the fourth-order fit led to regions of SLR greater than 14 cm within the domain. Improving this fit is a potential avenue of future research.

Changes: We mention the reason for choosing the third-order fit and that other fits were explored in the revised manuscript (Page 14, Line 11).

*P12, L24-25: The authors state that the shift towards higher transient climate response is driven by higher climate sensitivity in MESM, but there is not enough explanation in my opinion as to why there is a larger CS in MESM compared to previous versions, how they compare (e.g. posterior distributions), and to what extent the updated forcings play a role.*

Response: Through the points made to previous comments and the changes made to the manuscript, we believe that this has been more clearly addressed. Looking at the response surfaces, for any Kv value, an increase in ECS leads to larger TCR. Thus, given a constant Kv distribution, shifts towards higher ECS result in a shift towards higher TCR. With the relatively small changes in the Kv distribution from the subsampled MESM ensemble (see Table 2 of the manuscript), we find the assumption of constant Kv distribution needed for this argument justifiable.

Changes: As mentioned in the responses/changes to the comments above, we have added a discussion for each parameter that explains how changes to the model forcings can lead to the shifts observed in the marginal distributions. Furthermore, the addition of Figure 5 provides a direct comparison of the posterior distributions for each parameter derived from IGSM and MESM.

*P12-13: The conclusions provide a nice summary of the paper's key points. I suggest expanding the results section to include more in-depth discussion along these lines.*

Response: As noted above, we have expanded the results section to provide more in-depth discussions of the reasons for the changes in the parameter and TCR estimates.

Changes: Specific changes to the results section (Section 4) are given in responses to earlier comments.

**Response to Reviewer #2**

*1) I miss a description of the basic components and parameterizations of the model in the method section 3. I miss a section that describes model spin-up and the setup for the different model simulations, including external forcing factors. Further, it is not evident from the description why the model is called "Earth System Model". For example, are biogeochemical cycles included? Does dynamic vegetation affect albedo? Is it an ESM or rather an Earth System Model of Intermediate Complexity? I also miss a brief description of the metric used to compare model and data and how they are used to derive probability distribution. It is not sufficient to refer the reader to the literature (Libardoni and Forest 2011).*

Response: The MIT Earth System Model is an integrated model with sub-models for the atmosphere, ocean, land surface, atmospheric chemistry, ocean biogeochemistry, and the terrestrial ecosystem. When all of these sub-models are turned on, the model is set up as an Earth system model. However, under that set up, the model is too computationally expensive to be used for probabilistic studies of the model parameters like what is presented in this study. Turning off all components of the model except the atmospheric, ocean, and land surface models simplifies the model to an EMIC that can be used for probabilistic estimates of the model parameters investigated in this work.

Changes: A more detailed presentation of the EMIC (climate component of MESM) has been added to Section 2. In that discussion, we describe the model components of the EMIC, the input forcings, and the model parameters. In the discussion of the model parameters, we describe how each of the three are adjusted and how the model is being modified to make the changes.

In Section 3, we have included a summary of the methods used to derive the probability distributions. We present the goodness-of-fit statistic used to evaluate the model. This statistic is the weighted sum-of-square residual between the model output and observed climate record for a given diagnostic. A reference to the likelihood function is provided and we explain how the joint distribution is calculated from the goodness-of-fit statistic.

*2) Section 3: The authors vary three parameters – ocean diffusivity, an aerosol forcing scaling, and the strength of the cloud feedback determining ECS and constrain the models with two parameters.*

*2a) There is little information in the method section what these parameters specifically influence. The aerosol forcing scaling is unclear. Does this mean that all aerosol forcings are lumped together and scaled with a constant time invariant factor? How are different uncertainties applying to different aerosol classes (e.g. sulfate versus soot) considered or not and what is the justification for this approach. Please discuss caveats related to your assumption of a scaling factor.*

Response:  In the description of the model parameters that was added to Section 2, we describe what each of the parameters influence.  For completeness, we summarize them again here.  ECS is modified by adjusting the strength of the net cloud feedback in the model.  More specifically, a number of simulations where $CO_2$ concentrations have been doubled and the system brought to equilibrium have been run for different values of the cloud adjustment.  These are used to provide a lookup table which gives the cloud adjustment needed for a specific ECS.  Ocean diffusivity is defined by a latitude-dependent pattern based off of tritium mixing into the deep ocean.  Kv represents the global mean value and specific diffusivity values are calculated by scaling the spatial pattern by the same value at all latitudes to achieve the desired global mean value.

The forcing due to all aerosols except sulfate are held constant during historical simulations and the sulfate aerosol is parameterized through adjustments to the surface albedo based on changes in the historical emissions of $SO_2$.  The historical emissions have both spatial and temporal components, with the aerosol parameter setting the amplitude of the pattern in the 1980s.  Adjusting the forcing in this manner is not without its drawbacks.  As the only adjustable forcing component in the model, this forcing pattern also represents an estimate of all other forcings not included in the model.  Thus, this is not a pure estimate of the aerosol forcing.

Changes:  We have added a description of the model parameters, what they represent, and how they are adjusted into Section 2.

*2b) Effective ocean diffusivity is a very loose term. Is this diapycnal, vertical or horizontal diffusivity or does the parameter refer to the diffusivity associated with Gent-McWilliams parameterization? The subscript v of Kv points to vertical diffusivity. I would hope that this parameter reflects diapycnal diffusivity as diapycnal diffusivity co- governs ocean overturning strength and thus surface-to-deep heat transport. In any case, I am puzzled about the range sampled. Diapycnal diffusivity in coarse resolution, dynamic ocean models is typically of order 0.1 10-4 m2 s-1. Here diffusivity is varied in steps of 1 10-4 m2 s-1 and a very wide range up to 64 10-4 m2 s-1 is used. The upper value is even much larger than applied in classical box-diffusion models (1-2 10-4 m2 s-1 ); in box-diffusion models the entire vertical transport (mixing, advection, convection) is parameterized by diffusion only. What is the justification for this large sampling range? As a minor point, please use SI units for diffusivity. Further, I though Gent-McWilliams parameterization is included in the MIT model. If yes, why is the Gent-McWilliams diffusivity not varied or is this parameter linked with the "effective diffusivity"?*

Response:  In the ocean model, horizontal heat transport is prescribed by the Q-flux calculation and the vertical mixing of heat into the deep ocean is prescribed by the spatial diffusivity pattern and scaled by Kv as discussed above.  As Kv represents the mixing of heat into the deep ocean by all processes, it is greater than diapycnal diffusion values found in the sub-grid scale parameterizations of dynamic ocean models.

A wide range of Kv values was sampled to simulate many possible climate states, including those with very strong vertical ocean mixing.  Similarly, wide ranges were also chosen for

climate sensitivity and the aerosol forcing.  For the most part, runs with extreme values of any parameter were rejected for being inconsistent with the model diagnostics.  In the case of Kv, this supports the claim that such high values should not have been sampled to begin with.  The penalty paid for this over sampling of the parameter ranges is a misallocation of computing resources.

Changes:  We have added text to the manuscript in Section 2 to address these concerns.  We have clarified that a mixed-layer ocean model is used, that Kv represents the mixing due to all processes, and how the mixing is spatially distributed.

*2c) ECS is typically used to abbreviate Equilibrium Climate Sensitivity. Here, an effective climate sensitivity is introduced and termed ECS. What represents this effective climate sensitivity?*

Response:  We mistakenly expressed ECS as effective climate sensitivity, when it is, in fact, equilibrium climate sensitivity.  The lookup table for ECS is derived from runs brought to equilibrium, so that any equilibrium climate sensitivity can be obtained through the proper adjustment of the cloud feedback.

Changes:  All references to effective climate sensitivity have been changed to equilibrium climate sensitivity.

*3) Section 3: I question somewhat the application of only two observational metrics to constrain ECS, TCR, and sea level rise. Namely, pattern of surface air temperature change and "linear" ocean heat uptake are used as constraints by the authors. In my opinion, there is a lack of observational constraints to probe the timescales of deep ocean overturning (e.g. 14C). Thus it appears not surprising that the diffusivity parameter remains not well constrained. There is also a lack of metrics to probe the spatial pattern of heat uptake. This is particularly important as the thermal expansion coefficient varies by almost an order of magnitude in the ocean. Thus it matters, where the heat is taken up to estimate sea level rise. As another focus of the study is on TCR, it would also be nice to invoke additional metrics on thermocline ventilation as for example available by observation-derived fields of CFCs and bomb-produced 14C.*

Response:  Given the mixed-layer ocean model that is coupled to the atmosphere, we are somewhat limited to the diagnostics that can be used to evaluate the ocean system.  As further explained above, the vertical mixing pattern is prescribed with latitudinal dependence, but also fixed throughout the run.  The vertically-integrated horizontal heat transport is also prescribed based on offline Q-flux calculation.  With these patterns fixed, incorporating ocean diagnostics with spatial dependence is not feasible at this time.

As an aside, developing additional model diagnostics to constrain estimates of the model parameters, TCR, and sea level rise is a task that should be undertaken and is of interest to the authors.  Care should be taken to ensure that these metrics are independent of each other or that steps be taken to account for the correlation between metrics.  However, developing such metrics is beyond the scope of this work.

Changes:  We have added a discussion to Section 3 that explains why we chose the model diagnostics that we did (Page 5, Line 24).  We include references to other work that helps justify our choices.

*4) Page 5 to page 7, results, The description of the difference in input forcing is useful, but in my opinion misplaced. Solar and ozone forcings are model drivers (or forcings) and distinct from a particular model version. These forcings should be described in the method section where the simulations and the applied external forcings are to be described.*

Response:  While we recognize that the presentation of the model forcings may be better placed in the methods section, we believe that keeping them in the results section is justifiable. The interpretation of the new forcings and their direct application to the model parameters are in themselves a finding in this study.  Much of the reasoning for the shifts in the parameter estimates centers around these changes in the model forcings and are essential to the explanation of the results.  In our opinion, keeping them together is appropriate.

Changes:  No major changes have been made to the location of this discussion.  However, we have further clarified that only time variant changes to the forcings impact the historical simulations (Page 7, Line 4).

*5) P6, line 3ff; Q-flux adjustment: Does this mean that the authors apply temperature flux correction to their model? This should be explained in the method section.*

Response:  We have addressed the Q-flux adjustment, which represents the vertically-integrated heat flux, earlier in this response.

Changes:  An explanation of the Q-flux adjustment has been added to the manuscript and discusses how it is related to horizontal heat transport in the ocean (see Section 2).

*6) Section 4: I miss a figure comparing the modelled pattern of the median (or mean or best-guess version) with the observed pattern of surface air temperature change and similar for the global ocean heat uptake and its spatial pattern (and may be for upper air temperature) to illustrate how well the model is able to capture the observations.*

Response:  A figure comparing the model output to the observed surface pattern used in our diagnostic does not yield a clean comparison.  As a result of weighting the model-to-observation residuals by the noise covariance matrix, the temperature patterns are rotated into a coordinate space defined by a set of orthogonal basis functions defined by the internal variability estimate.  Thus, any attempt to compare the model output and observations in the unrotated space does not give a fair representation of an individual model run's fit to the observed record.  A fairer assessment of the model fit to the observations is obtained by comparing the global mean temperature time series.

Given the fixed mixing pattern used in the ocean model, the spatial pattern of heat uptake does not vary between the model simulations. Only the magnitude changes, making a comparison between the model and observations for individual runs redundant.

Changes: We have included a figure where the global mean surface temperature of each of the 1800 model runs is shown, along with the observed time series for each of the five datasets used in this study. We have also highlighted the model runs where the parameter settings most closely match the median values from the marginal distributions derived from each of the surface datasets. All anomalies are calculated based off of the 1906-1995 climatology used in the surface diagnostic.

Similar to the global mean surface temperature results, we also include a figure to show the spread in the ocean heat content linear trends calculated from our ensemble. We plot a histogram of the calculated trends from each individual run, while also showing the observed trend and highlighting the runs with parameter settings closest to the distribution medians.

A discussion of these results begins on Page 12, Line 11. In this discussion, we explain the model's ability to match the observations and reasons for mismatches between the model output and the historical record.

*7) Page 12, line 7: How well does the polynomial fit represent the model results?*

Response: In general, the polynomial fit represents the model results quite well, but is not without error. In our response to Reviewer #1, we discussed using first-, second-, and fourth-order fits, as well as some of the errors associated with the third-order fit.

Changes: We mention the reason for choosing the third-order fit and that other fits were explored in the revised manuscript (Page 14, Line 11).

*8) Page 12, line 14: Why is the PDF for the TCR not directly estimated from the 372-member ensemble? Does the fitting add additional uncertainties to the procedure of estimating TCR?*

Response: It is possible to directly estimate the PDF for TCR from the 372-member ensemble. Doing such would represent estimating TCR from a joint distribution where all values of ECS and Kv are equally likely to occur. In other terms, the ECS-Kv two-dimensional PDF would be uniform for all pairs within their respective domains. We have shown in this study that ECS and Kv are not uniformly distributed and that some pairs are more likely to occur than others. Drawing from this more realistic distribution yields a probability-weighted sampling of parameter pairs from which to estimate TCR.

Using the polynomial fit adds additional uncertainty to the procedure of estimating TCR by introducing interpolation error. As described in the response to Reviewer #1, the polynomial fit is not an exact match to the model results, and any error in the estimation propagates as an error in the TCR distribution. However, running the transient simulation for each ECS-Kv draw

from the Latin Hypercube Sample is infeasible, so the fit is required to estimate TCR for the pairs where there is no corresponding run.

Changes:  No major changes have been made to the manuscript.

*9) Discussion and conclusion: While the authors suggest that their approach should serve as a template for other groups, they fail to mention that similar, and sometime much more comprehensive approaches of parameter calibration, have been undertaken by other groups. They also fail to compare their estimate of TCR and ECS with published estimate and to put their findings in the context of the wider literature. See for example, Collins et al., IPCC, 2013 for the most recent assessment of TCR and ECS values by IPCC. Of course there are recent updates of these estimates and there are also many other studies that determine model parameters such as vertical ocean diffusivity. Examples that come immediately in my mind are Holden et al., Clim. Dyn., 2010, Richardson; Nat. Clim.Change, 2016, Schmittner et al., GBC, 2009, Steinacher et al., Science, 2013 or Steinacher and Joos, Biogeosciences 2016. It is the task of the authors to identify the recent literature to provide a relevant discussion.*

Response:  In both the abstract and the penultimate paragraph of the introduction, we state that the point of the study is to assess how the changes in the model can impact the distributions.  The paper is not intended to discuss how the results compare with recent estimates of ECS or TCR distributions or specific methodologies for estimating probability distributions.   We think the introduction's text reflects this and is included here.

" In this study, we provide a transparent method of testing and accounting for how the simulated behavior and probability distribution functions change in response to the recent model development. We derive a new joint probability distribution by closely following the methods of Libardoni and Forest (2011) to show the impact that the new version of the model has on the parameter estimates and find that the new version of the model leads to higher climate sensitivity estimates in addition to shifts in the distributions of the other model parameters. The effects on the parameter distributions due to changing observations and temperature metrics will be addressed in future studies in order to separate their impacts from changes due to the model update alone"

The future work will provide the appropriate discussion of other studies as suggested by the reviewer while this work only documents the impact of changes in the model framework.

We are aware that other approaches exist and have avoided stating that our parameter estimation methodology is better.  We do think this approach can serve as a template for testing how new versions of models can directly impact parameter estimates and that such tests should be documented in a similar fashion.

Changes:  Throughout the manuscript, we have made it clearer that we are only conducting the baseline test of the model in this study.  Examples include Page 3, Line 3 and Page 12, Line 11 of the revised manuscript.  These areas defer discussion of results that don't compare the IGSM estimates to the MESM estimates to future/concurrent work.

*P1, Line 22: typo: sensitivity*

Changes:  We have fixed this typo in the manuscript.

[revised manuscript text omitted]